# What Pediatricians Should Know before Studying Gut Microbiota

**DOI:** 10.3390/jcm8081206

**Published:** 2019-08-12

**Authors:** Lorenzo Drago, Simona Panelli, Claudio Bandi, Gianvincenzo Zuccotti, Matteo Perini, Enza D’Auria

**Affiliations:** 1Department of Biomedical Sciences for Health, Università di Milano, 20133 Milan, Italy; 2Department of Biomedical and Clinical Sciences “L. Sacco”, Pediatric Clinical Research Center “Invernizzi”, Università di Milano, 20157 Milan, Italy; 3Department of Biosciences, Università di Milano, 20133 Milan, Italy; 4Department of Pediatrics, Children’s Hospital Vittore Buzzi, Università di Milan, 20141 Milan, Italy

**Keywords:** gut microbiota, microbiome, maternal–fetal interface, newborn, child, pediatric disease, dysbiosis

## Abstract

Billions of microorganisms, or “microbiota”, inhabit the gut and affect its homeostasis, influencing, and sometimes causing if altered, a multitude of diseases. The genomes of the microbes that form the gut ecosystem should be summed to the human genome to form the hologenome due to their influence on human physiology; hence the term “microbiome” is commonly used to refer to the genetic make-up and gene–gene interactions of microbes. This review attempts to provide insight into this recently discovered vital organ of the human body, which has yet to be fully explored. We herein discuss the rhythm and shaping of the microbiome at birth and during the first years leading up to adolescence. Furthermore, important issues to consider for conducting a reliable microbiome study including study design, inclusion/exclusion criteria, sample collection, storage, and variability of different sampling methods as well as the basic terminology of molecular approaches, data analysis, and clinical interpretation of results are addressed. This basic knowledge aims to provide the pediatricians with a key tool to avoid data dispersion and pitfalls during child microbiota study.

## 1. Introduction

The field of microbiome research is quickly evolving and unravelling. Causal links between distinct microbial consortia, their collective functions, and host pathophysiology during the various stages of life are becoming increasingly clear. Studies of microbiome plasticity, composition, and function based on a distinction of the host phenotypes may lay the foundation for both therapeutic and preventive interventions [1]. Indeed, new practical aspects of microbiome studies will be focused on the personalization of actions as well as on an understanding of the inherent individual variability of microbiomes at different ages, stages of development, conditions, and internal or external influences. These studies will allow the comprehension of physiological features to explain, or predict, human health and disease states. Therefore, clinical studies need to be well designed and the subject/patient phenotype properly selected. Age and many other factors have the potential to strongly influence the results, thus clinical studies on microbiota in children should take into account the differences that naturally occur during growth. Other technical challenges that need to be addressed are linked to properly establishing, harmonizing, and standardizing clinical protocols for sample collection, processing, sequencing, and analysis that also takes into account the “microbiome’s age”. The issues of diet, environment, host immune system, and genetics as key factors for determining microbiome and microbiota profiles have not been fully resolved yet. All of these influences can impact on the microbiota composition at any age and may sometimes be difficult to harmonize and standardize during clinical investigation.

Clinical and microbiological translation urgently needs to implement the main information on microbiota. This review aims to give a rapid overview of child microbiota in order to guide pediatricians to a better understanding of the field while trying to limit biases and intrinsic pitfalls before the study design and starting any clinical trials. Even if most of the reported literature and data specifically refer to the best studied community, in other words, the one inhabiting the gut, the knowledge discussed in the text, together with more practical aspects and recommendations, can also be adapted to the study of other medically-relevant communities (e.g., in nasal-oral cavities).

## 2. Basic Knowledge on Gut Microbes

The human body harbors trillions of microbial cells mainly represented by bacteria, but also includes archea, viruses, fungi, and parasites. These communities establish extensive networks of cross-feeding (trophic) interactions, consuming, producing, and exchanging hundreds of metabolites with each other and with their human host, with whom they constitute a unique ecological entity called “holobiont” [2,3]. Their highest density is reached in the intestinal compartment, particularly in the lower segments. Here, bacteria are estimated to reach a number of 10^14^ cells and their density in stool have been calculated in the order of 10^11^ per gram of dry material [4]. Although less-well studied, many other body habitats within healthy individuals are occupied by microbial communities such as the mouth and oral tract, nostrils, skin, vagina. The term ‘microbiota’ literally means all living organisms within a body-site habitat. More specifically, the term “gut microbiota” indicates the resident intestinal bacterial communities, and from a practical point of view, it is generally investigated, with obvious biases, through the analysis of fecal samples, which are easy and non-invasive to collect. The term ‘microbiome’ is used instead to refer to the genetic content of these microorganisms. Conventionally, research in the field is mainly focused on bacterial microbiome, but further fascinating results have come from the study of “virome”, or the viruses inhabiting the gut, of “mycome”, which reveals another intriguing world of gut fungi, and of “parasitome”.

New genetic and sequencing technologies have opened the way to the ‘metagenomic’ approach, which directly analyzes the total microbial genomes contained in a sample, that in turn, allows information to be acquired on the genomic links between function and phylogenetic evolution. Other approaches faced in the field include ‘metatranscriptomics’, the study of the whole RNA repertoire from a microbial community; ‘metaproteomics’, the study of the entire protein content from the community; and ‘meta-metabolomics’, the study of small-molecule metabolites produced through the interaction of diet and microbiome [5,6,7].

The analysis of the gene coding for the ribosomal 16S rRNA is very useful for studying gut bacteria. 16S rRNA is a component of the prokaryotic ribosome and is coded by a gene spanning about 1500 bp. The 16S rRNA gene is highly conserved between different species of bacteria, but presents nine variable (“V”) regions that allow identification at the genus or species level. After amplification of, typically, 2–3 V regions, the obtained sequences are clustered into nearly-identical tags called ‘phylotypes’ or ‘operational taxonomic units’ (OTUs). These terms refer to a group of microbes generally through the threshold of sequence homology between their 16S rRNA genes (e.g., ≥98% for a ‘species’-level phylotype) [8].

Eukaryotic components of the microbiota (e.g., fungi and protozoans) can be analyzed through homologous ribosomal gene sequences (small-subunit rRNA, SSU rRNA), while viral communities that lack ribosomal genes are investigated through shotgun DNA sequencing, or via primers targeted on conserved sequences in viral families. The above approaches are referred to as culture-independent, while culturomics is a culturing approach that uses multiple culture conditions, combined with the MALDI-TOF mass spectrometry and/or the 16S rRNA sequencing, for the isolation and identification of the largest possible number of bacterial species [9]. 

The gut hosts taxonomically diverse archaea, bacteria, fungi, and viruses. Studies report at least 22 bacterial phyla in the body, mainly represented (>90%) by *Actinobacteria*, *Firmicutes*, *Proteobacteria*, and *Bacteroidetes*. In the gut, *Bacteroidetes* and *Firmicutes* represent the predominant phyla [10,11,12]. In addition to taxonomic composition, taxonomic diversity also needs to be considered in evaluating the homeostasis of microbiota. In particular, two parameters are routinely employed for this purpose: alpha diversity (within-sample diversity, how many taxa or lineages are present in a sample), and beta diversity (between-sample diversity, to which extent the guts of different subjects or patients share taxa or lineages). Parameters that need to be evaluated when computing these ecological indices are richness (i.e., how many bacterial taxa) and evenness, which also takes into account the relative abundance of taxa, in addition to presence/absence, and compares it between subjects or patients [13].

In this context, measures of species richness (for example, the number of observed species or the Chao1 index, which is an abundance-based estimator of species diversity) and phylogenetic measures (Faith’s phylogenetic diversity) are sensitive to the number of sequences per sample, whereas this is true to a much minor extent for metrics that combine richness and evenness (Shannon index).

Statistical and computational analyses still remain the main challenge in microbiome research. Some methods currently used for their power and effect size analysis are based on PERMANOVA, Dirichlet Multinomial, or random forest analysis [14]. Parametric statistical tests (for example, the Student’s t-test and ANOVA) as well as measures of correlation including Spearman’s rank correlation can be used on the basis of the phenotypes under study and the type of information the researcher wants to capture.

## 3. The Intestinal Microbiota from Birth Throughout Childhood

Addressing neonatal and early-life microbiota is pivotal as many of the events capable of shaping microbial communities even in adults take place during this phase of life: gestational age at birth, type of delivery, breast vs. formula feeding, weaning, use of antibiotics, etc. [15,16]. When neonatal microbiota begins is still a subject of great debate. The “sterile womb paradigm”, in other words, the notion that, under physiological conditions, the human fetal environment is sterile and microbial colonization begins with birth, has been accepted for decades. Recently, with the burst of metagenomic studies, there has been a group of papers that have found traces of a lowly abundant bacterial colonization in the placenta, endometrium, amniotic fluid, and meconium in healthy, full-term pregnancies (see Nature Editorial by C. Willyard, 2018, [17] and references therein). This has led some researchers to date back the seeding of the microbiota to before birth (“in utero colonization hypothesis”). The field is still the subject of much debate, and the results appear in general to be controversial. Recently, several scientists have underlined that, even if it is possible that not all healthy babies are born sterile as previously thought, particular caution is necessary when working on samples bearing a low microbial biomass due to the heavy contamination issues notoriously connaturated with such samples when using molecular approaches based on next-generation sequencing [17]. Other important points that have been raised are the difficulty of maintaining a strict sterility when collecting samples related to the in utero environment within a clinical setting, and the impossibility of using NGS-based techniques to discriminate DNA from viable cells and DNA belonging to dead organisms or derived from translocation from the blood stream [15,17]. 

The human intestine at birth is an aerobic environment, as such, while the adult gut microbiota is dominated by obligate anaerobes belonging to the *Firmicutes* and *Bacteroidetes* phyla, the neonatal pioneer flora is composed by aerotolerant taxa, mainly belonging to the *Enterobacteriaceae* family (phylum: *Proteobacteria*). In a matter of days, however, these microorganisms will reduce oxygen levels, and the intestinal lumen becomes anaerobic. This allows the colonization by strict anaerobes, dominated by *Bifidobacterium* (phylum: *Actinobacteria*); *Clostridium* (phylum: *Firmicutes*); and *Bacteroides* (phylum: *Bacteroidetes*) [18,19]. During the first months, the diet of the infant is almost exclusively milk, favoring milk oligosaccharide fermenters as the already cited *Bifidobacterium*, represented, at this stage, by many species. Other predominant bacterial taxa are represented by *Enterococcaceae*, *Streptococcaceae*, and *Lactobacillaceae* [15].

A very recent paper [20] addressed the development of gut microbiota in a large cohort of children, comprising cases who seroconverted to islet cell autoantibody positivity, children who developed type 1 diabetes (T1D), and matched controls (healthy). This interesting analysis followed the longitudinal maturation of the microbiome from 3 to 46 months of age and determined the covariates that significantly affected its development. Globally, this study harmonized data by collecting 12,500 stool samples from 903 children in three different European countries and three US states. Breastfeeding and birth mode resulted in being the main factors able to drive gut microbiome during the developmental phase by changing some relevant bacterial clusters. The authors proposed three distinct phases of microbiome progression: a developmental phase (months 3–14), a transitional phase (months 15–30), and a stable phase (≥31 months). The Shannon diversity index changed significantly during the first two phases, unchanging only during the stable phase. This study represents a very nice model of how to harmonize the age of the children with other covariate factors. Figure 1 presents a proposal for pediatricians to use a personalized staging of the enrolled individuals to differentiate relevant microbial clusters and dominating phyla. 

## 4. Issues to be Considered for Studying Microbiome in Clinical Studies

### Study Design and Patient Selections

Pediatricians should select children cohorts by trying to limit the confounding factors that have the potential of diluting the statistical estimates of the effect sizes of the microbiome. Thus, as an example, when defining disease-specific signatures, the diseased population should be recruited with particular care in choosing patients who display a relatively homogeneous clinical phenotype. The choice of controls is also a challenging question: a good control population includes patients with a clinical phenotype that is a clear contrast from the one under study, while matching other relevant criteria. To reduce the heterogeneity of the cohort, it is indeed mandatory to clearly define inclusion and exclusion criteria by considering the factors affecting microbiota analysis (see below) and matching, accordingly, cases and controls. In this regard, it is crucially important to collect information about potential confounding factors, among which age group, for moderating influences that can artifactually alter results and the outcomes of interest. This is important in order to decrease co-variability and heterogeneity during the enrollment, by increasing the power of the analysis in parallel. The collected information will form part of the “metadata” (covariates) surrounding the sample and will later be used in analyzing the data. To ensure consistency, recording the maximum information about the subjects, sample, and experimental procedures is recommended. Finally, before starting the study protocol, a sample size should be estimated on the basis of the expected effect size, and evaluated by means of a pilot study or based on similar previous studies. Other recent approaches rely on computing the estimated sample sizes by calculating the independent effect sizes on microbiota variation of other factors (covariates) relevant to the phenomenon under study [21]. 

Table 1 summarizes the key aspects to consider when designing and conducting a microbiome study, lists the possible confounders and pitfalls, and presents practical solutions for risk mitigation.

## 5. Major Pre-, Peri-, and Post-Natal Factors Affecting the Child Gut Microbiota

A schematic representation of the factors that are able to affect the dynamics and composition of the intestinal microbiota is given in Figure 2. 

### 5.1. Maternal Factors Influencing Infant Microbiota

#### 5.1.1. Changes Related to Vertical Transmission of Maternal Metabolites

During gestation, bacteria in the mother’s intestine have been shown to drive the future immune maturation of the neonatal gut through the passage of soluble molecules from the placenta in the absence of direct colonization and of the vertical transmission of viable bacterial cells [22,23]. These bacteria are able to induce specific changes in the gut of newborns, creating new microbiota profiles. 

#### 5.1.2. Changes Related to Dietary Patterns and Lifestyle

The intestinal microbiota is strongly personalized and influenced by a plethora of environmental and inter-individual variables including body mass index (BMI), exercise frequency, and dietary patterns and habits (which in turn, are strongly related with cultural factors and lifestyle). It has been reported that the infant’s fecal microbiota composition is influenced by the BMI and weight gain of the mother during pregnancy [24,25]. In general, the maternal microbial reservoir plays a crucial role in the acquisition and development of early infant microbiota, which in turn is the key to establishing a healthy host–microbiome symbiosis with long-lasting health effects. Therefore, it can be easily understood as to why maternal diet and lifestyle should be monitored and categorized as relevant metadata in infant microbiota studies. In an early phase, after the huge microbial “inoculum” at birth, the infant continues to directly acquire maternal gut strains from different sources (e.g., from skin, mouth, milk) and these are likely to become stable colonizers of the infant gut. Later in life, increasingly important roles are also played by other factors such as shared diet and lifestyle.

### 5.2. Genetic Factors

There is growing evidence that geographical origin and host genetic makeup influence the acquisition and development of the gut microbiota, with clear associations reported between the host genotype and the relative abundances of different bacterial taxa. For example, Bonder et al. [26] described a single nucleotide polymorphism (SNP) in the LCT locus (coding for human lactase) that is related to varying abundances of *Bifidobacterium*. Goodrich et al. [27], by comparing microbiota across samples belonging to either monozygotic and dizygotic twin pairs, reported a number of microbial taxa whose abundances were strongly influenced by host genetics. Among such taxa, the *Christensellaceae*, considered a microbiome-based marker of obesity and is significantly enriched in individuals with low BMI, resulted in the most highly heritable taxon. Any data related to the genetic hardware of the child should then be noticed.

### 5.3. Mode of Delivery

At birth, the infant gut communities tend to resemble the maternal vagina or skin microbiota in cases of vaginal or cesarean section (C-section) delivery, respectively [19,28]. Even later, when these “pioneer” foundation populations have been replaced, the birth mode seems to exert significant long-term effects on the structure of the gut microbiota. At 24 months of age, the gut microbial communities of cesarean delivered infants still appear to be less diverse [15]. Even in children as old as seven years, some authors have reported the enduring influence of the mode of delivery, but data are somewhat contrasting regarding this point [19]. Vaginally delivered infants tend to be colonized by *Lactobacillus* and *Prevotella*, while C-section neonates are preferentially colonized by microorganisms from maternal skin, and the hospital staff or environment.

### 5.4. Mode of Infant Feeding

Breastfed infants receive, from their mothers’ milk, a complex mix that will affect the milieu within which their own microbiota will develop. This mix is made up of nutrients, antimicrobial proteins, short chain fatty acids (SCFA), secretory IgA, non-digestible oligosaccharides (HMOs, human milk oligosaccharides, that promote the proliferation of specific gut bacterial taxa in the neonate), and live bacteria, even if previously considered germ-free [15]. The source of the “milk microbiota”, which has a transient nature and declines rapidly at weaning, has recently been another subject of debate. At least some of the bacteria is thought to reach the mammary gland through an endogenous route called the enteromammary pathway, which has not been fully elucidated yet. It has also been suggested that mammary skin microbiota can travel via the lymphatic and vascular circulations to the breast ([15,16] and references therein). Gut microbiota differences between breastfed and formula-fed infants are indeed well documented. The former exhibit lower diversity indexes, indicative of a more uniform population where *Bifidobacterium* and *Lactobacillus* dominate. The latter are characterized by more diverse communities, with higher proportions of *Bacteroides*, *Clostridium*, *Streptococcus*, *Veillonella*, *Atopobium*, and *Enterobacteriaceae* [29]. Finally, compositional differences in microbial communities in human milk sampled from different geographical locations have been studied and reported to create strong variability between newborn microbiota [30].

### 5.5. Gestational Age

While in full-term infants, delivery and feeding mode are reported to represent the major drivers of microbiota development, in preterm (PT) infants (<37 weeks of gestation), the gestational age seems to have the biggest impact on the assembly of gut communities [19,31,32]. PT neonates experience a number of unique challenges in the establishment of their microbiota. Their colonization patterns are characterized by the involvement of peculiar microbial sources, mainly bacteria deriving from the neonatal intensive care unit (NICU) environment [33]. Not rarely, these are strains implicated in nosocomial infections such as *Enterococcus* spp., *Staphylococcus aureus*, *Klebsiella pneumoniae*, *Acinetobacter* spp., *Pesudomonas aeruginosa*, and other *Enterobacteriaceae* [34] with their burden of antibiotic resistance genes. Other relevant features of this peculiar colonization trajectory are its extreme inter-individual variability, and the fact that, across studies, it does not appear to be univocally linked to health outcomes as necrotizing enterocolitis and late-onset sepsis. Instead, the colonization process seems to reflect the co-occurrence of a variety of nosocomial “variables” [35], among which are parenteral nutrition and antibiotic usage (see below). Antibiotics, normally administered to these patients, in turn perturbate the colonization process by killing bacteria acquired during birth and promoting the growth of taxa significantly different from those found in more physiological situations [31]. In conclusion, the PT microbiota appears to be more unstable than that of full-term equivalents and is believed to be associated with a delay in the establishment of an adult-type signature microbiota [16]. All these individuals should be carefully selected and clearly categorized by the clinician before enrollment into the microbiota study.

### 5.6. Antibiotics

Specific properties of antibiotics, as a mode of action and antimicrobial spectrum, might act as powerful forces for the selection of intestinal bacterial populations, especially if the infant is exposed to antibiotics too early and/or for long periods of time [3,15]. Antibiotics are able to alter the abundances of resident bacteria, significantly impact the growth of otherwise dominant bacterial phyla, and lead to an overall decrease in microbial diversity. A study by Fouhy and colleagues [36] showed that infants exposed to ampicillin and gentamicin shortly after birth harbored higher proportions of *Proteobacteria* and *Actinobacteria*, and the genus *Lactobacillus* for up to four weeks after concluding treatment. Another study reported an attenuation in colonization with *Bifidobacterium* and an increase of *Enterococcus* in subjects receiving oral or intravenous antibiotics during the first four days of life [37].

This variability among individuals suggests caution when including subjects who have been treated with antibiotics [38]. Indeed, the exclusion criteria from the NIH Human Microbiome Project (HMP, dbGAP, see the url https://www.ncbi.nlm.nih.gov/projects/gap/cgi-bin/study.cgi?study_id=phs000228.v4.p1) include the use of systemic antibiotics, antifungals, antivirals, or antiparasitics within six months of sampling. However, this criterion, although optimal, may not be easily applicable with subjects in the pediatric age. For this reason, shorter time windows are often considered. In any case, it is mandatory to accurately document, within the metadata file, any history of antibiotics as well as other medication use.

### 5.7. Weaning

The transition to more varied, solid food is an important step in the development of the early-life gut microbiota; infants begin to be exposed to a much larger array of substrates and non-digestible carbohydrates that promote the survival and proliferation of more various bacterial taxa. As a consequence, the alpha diversity increases; moreover, *Proteobacteria* and *Actinobacteria* are replaced by *Firmicutes* and *Bacteroidetes* as the dominant phyla, in a more adult-like compositional structure. The cessation of exclusive milk feeding correlates with the decrease of saccharolytic bacteria as *Bifidobacteriaceae* (phylum: *Actinobacteria*). The increased protein intake is thought to be associated with an increase of *Lachnospiraceae* (phylum: *Firmicutes*), while the ingestion of fibers with that of higher levels of *Prevotellaceae* (phylum: *Bacteroidetes*) [39].

In general, the relative abundance of our intestinal microbes is highly influenced by dietary patterns and habits [11], that should therefore be taken into account in clinical studies targeting microbiota.

## 6. Minor Factors Affecting Gut Microbiota

Various minor factors can affect and modify the gut microbiota, which can occur at any stage of life. Insomnia and circadian rhythm disruption, latitude with time zone shift and intercontinental flights (with the consequent jet lag), household siblings, and companion animals as well as seasonal changes can modify gut microbiota and determine different microbiota profiles with high inter-individual variability to responses to the different factors [40,41,42]. All of these factors can influence the results and should be carefully considered before starting a clinical study and accurately reported in the metadata to then be considered later in the downstream bioinformatics and statistical analyses. Other similar confounder factors such as bowel movement preparations, evacuants or laxatives, or any microorganism-supplemented food (such as probiotics) can act as deep and long-time gut modifiers, thus a plot-to-plot variation needs to be addressed with nested statistical tests.

## 7. Sample Collection

Donors/patients to enroll, their genetic or disease phenotypes as well as the expertise of the clinician in methodology used for collecting samples are very relevant in designing a correct study. The number of samples and patients to be enrolled is an intriguing and still hotly debated topic. Sample stability as well as shipping and storage requirements need to be more appropriate and will surely be improved and standardized in the future. Researchers may find some procedures at http://www.microbiome-standards.org or at https://www.hmpdacc.org/resources/metagenomics_sequencing_analysis.php and other papers [43,44,45].

Concerning the practical aspects, an important question is how often to collect samples because the microbiome ecology is intrinsically dynamic. This largely depends on what question one is trying to answer. If, for gastrointestinal disorders, remarkable changes can be observed between one day and the next (e.g., in times surrounding surgery or in correspondence with periods of activity or remission of the pathology), changes induced by other factors (e.g., diet) often take place on a longer timescale. Collection of multiple samples from the same patient is preferred to allow for better standardization on the basis of the type of patients, centers involved, and statistical power. Whether or not samples collected from the same individual can be pooled before analysis is another topic to be standardized. An important point is that sampling and storage do affect microbiota composition in healthy as well as in diseased subjects. The most widely accepted protocols include immediate homogenization and freezing either with dry ice or in liquid nitrogen, followed by storage at −80 °C. However, this approach is not always practical, particularly for stool samples, or in the case of stool collection from a large scale cohort or remote/rural areas. Whether samples must be immediately frozen (and at what temperature) or whether they can withstand a period of room temperature remains controversial. The above-mentioned studies showed that the effects of short-term storage conditions on the structure and diversity of communities are quite small in general. In particular, storage at −80 °C, −20 °C for a week, or 4 °C for 24 h were found to not significantly affect the ecological indexes of between-sample diversity or the abundance of major taxa [45]. In contrast, the number of freeze–thaw cycles seems to have an effect on the composition of the microbial community, thus it is strongly recommended to aliquot samples at the beginning. Of course, some DNA stabilizers can be used to prolong the stability of samples. In the study of Choo et al. [46] Omnigene Gut and Tris EDTA appeared to show the same performance as storage in an ultrafreezer (−80 °C). In addition to feces, swabs can be an alternative starting material for DNA extractions, especially within hospital settings, even if some studies have shown that the stool swabs of some subjects had limited and not detectable bacterial DNA. A recent study by Christine M. Bassis [47], by comparing stool versus rectal swab samples and their storage conditions, demonstrated minor differences in the bacterial community profiles between the stool and swab from the same subject as well as when samples were stored up to 27 h at +4 °C before freezing at −80 °C. Interestingly, this study also concluded that it was possible to thaw and refreeze samples a limited number of times under particular conditions (i.e., immediately frozen at −20 °C, first thaw cycle, refrozen at −80 °C; immediately frozen at −20 °C, first thaw cycle, refrozen at −20 °C, second thaw cycle and frozen at −80 °C) without strong effects on the community composition. A word of caution is, however, due on this point, as the consensus recommendations are different, as detailed above. Finally, it is to be underlined that as the collection of stool can be difficult from some subjects under certain experimental conditions, swab collection may be useful in such cases, which also has the advantage that they are more easily shipped and handled. A further recent study confirmed that swab samples reliably replicate the stool microbiota bacterial composition when swabs are processed quickly (≤2 days) [48].

Finally, special considerations are needed if addressing peculiar samples such as the newborn’s first intestinal discharge (meconium). The debate about “when” the neonatal microbiota begins has been previously mentioned. Recently, several scientists have underlined that, even if it is possible that not all healthy babies are born sterile as previously thought, particular caution is due when working on samples bearing low microbial biomass such as meconia because of the contamination issues connaturated with molecular approaches based on PCR amplification and next-generation sequencing [17,49,50]. The presence of contaminating DNA in laboratory reagents (so-called “kitome”) is a serious challenge in these cases; low levels of target bacterial DNA in a sample have been reported to correlate with a high proportion of sequences being attributable to contamination [51,52].

## 8. Discussion

The Anna Karenina principle, based on Leo Tolstoy’s great book and cited in 1878 (*All happy families are alike: each unhappy family is unhappy in its own way*), has been recently translated by Zaneveld et al. [53] as the response to stress against the stability of animal microbiomes. These authors discussed how healthy microbiomes may be quite similar between individuals, but each dysbiotic microbiota is dysbiotic in its own way. The associations between microbiome instability/variability and many confounding factors as well as with diseases, suggest that microbiome may have many and simultaneous multiple faces.

This “stochastic” drift, occurring at any stage of life under stress conditions, can create several phenotypes that need to be known and harmonized when planning a study on microbiota.

Early childhood possesses distinct microbiota tracts compared with later ones, where different clusters and phyla may be differently represented. One common characteristic during this early stage of life is that bacterial richness and diversity increase during growth. Therefore, pediatricians should know that there are several age-related microbiota profiles, and should also be aware of the need to categorize each individual in a defined, monthly range by carefully considering the above-mentioned interference factors.

Several specialties need to be involved in this aim as well as the combination of different knowledge. The “Clinical Microbiota Expert” is not only a new job, but represents a step forward to create competence in this field where clinical microbiologists, clinicians, and bioinformaticians are merged into one. This new job-role will have to create awareness on the study of the “dynamic body” such as the gut microbiota during early age by creating novel models and approaches as well as solutions to solve and interpret the clinical microbiology results. Therefore, translational methodologies to approach a new way of designing clinical trials need to use feasibility and efficacy tools, and a deeper preparation in the field to avoid uncontrolled errors, unsubstantiated results, and excessive costs.

## 9. Conclusions and Future Perspectives

Next-generation sequencing methodologies still remain expensive and the diagnostic market is offering different solutions, thus a proper, and especially judicious, use of these methods is definitively mandatory. The clinical microbiota expert and pediatricians involved in the field will also have to guide through this jungle by trying to avoid false myths and promises that could be difficult to realize. In the near future, all of these studies and experiences will necessarily lead to a better understanding of the real key phases of microbiome progression from birth throughout childhood.

A final consideration to underline is that the metagenomics community still needs to fully converge toward standardized methods and procedures, leading to an investigation of the sources of variability and bias at each step of the workflow, and to an improved reproducibility and comparability between studies. This is a necessary premise for moving from correlation studies to causation investigations and to answer complex questions in a translational setting.

## Figures and Tables

**Figure 1 jcm-08-01206-f001:**
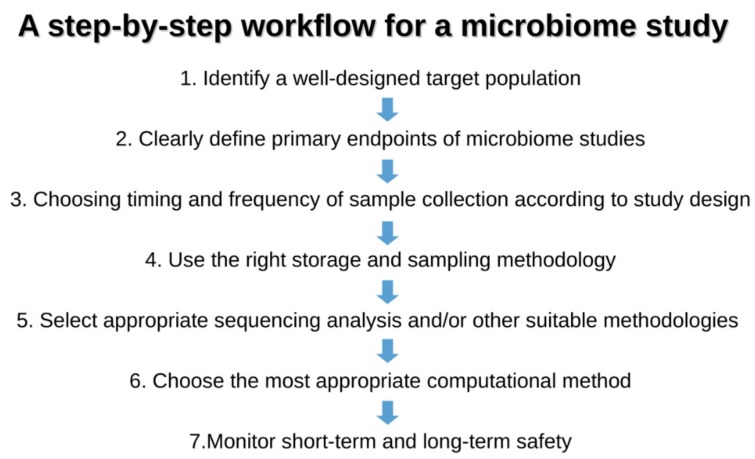
The figure represents the seven golden steps that the pediatrician should follow before the enrollment of individuals/patients in the microbiota study.

**Figure 2 jcm-08-01206-f002:**
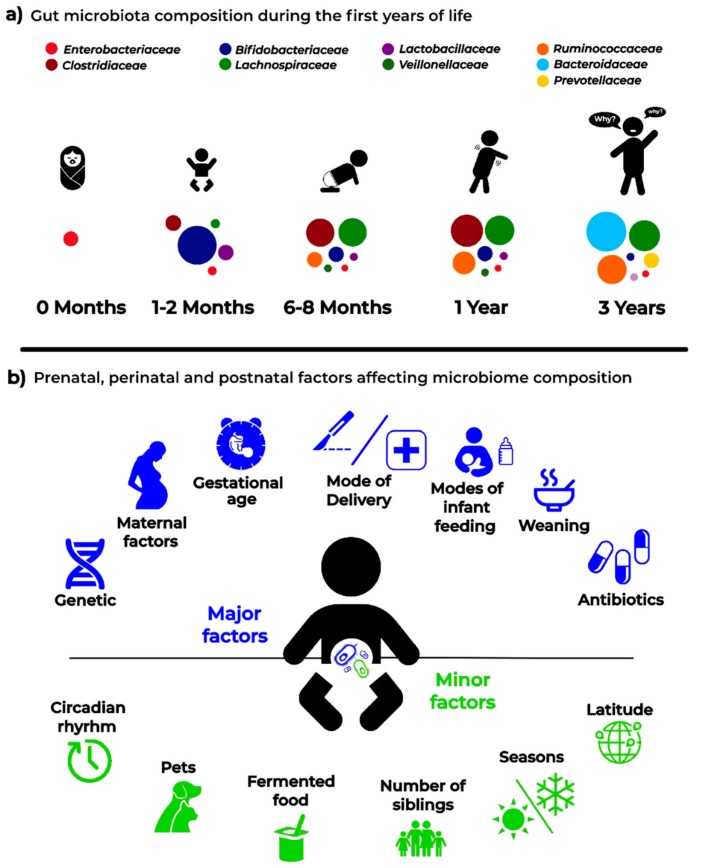
Infant microbiota composition (**a**) and the main “major” and “minor” factors affecting analysis and results in microbiota studies (**b**).

**Table 1 jcm-08-01206-t001:** Practical aspects to follow when drawing and studying a Microbiome.

Stages and Pitfalls	Considerations and Practical Solutions
Study question	Clearly define the aim(s) of the study and the relevant biological question(s) before setting up the study design.
Statistically underpowered studies	Correctly determine the sample size: consider that enrolling enough participants is important to ensure that the expected effect will be detected. The sample size can be estimated by means of pilot studies, or from previous similar studies, or alternatively from computational approaches that consider the effect of covariates on the total microbiota variation (see main text).
Selection of subjects: avoiding heterogeneity of the population	Clearly define inclusion and exclusion criteria: consider that an initial heterogeneity of the population will then dilute the statistical estimates of effect sizes on the microbiome. The list of exclusion criteria from the National Institutes of Health (NIH) Human Microbiome Project can be relied on with regard to the above-mentioned. In a “cases vs. controls” study, aimed at detecting microbiota-based markers of a disease, choose “cases” with a care in maintaining a relatively homogeneous clinical phenotype. “Controls”, in turn, must have a clinical phenotype in clear contrast, while matching other relevant criteria to avoid confounding factors. Consider that multiple controls groups that are selected based on various criteria may provide more insights. Additionally consider that for more generalizable results, independent cohorts may be selected to identify the microbiota signatures (“discovery cohort”) and test the results (“validation cohort”). In longitudinal studies, individuals can be treated as their own controls, by collecting baseline samples before and during/after a treatment.
Confounding factors (lifestyle and clinical factors)	Be exhaustive in the collection of “metadata” (covariates) surrounding the sample: this will be pivotal later, when analyzing the data. Collect information on possible confounding, mediating, and moderating factors that can either influence the microbiome composition or the outcome of interest.
Timing and frequency of sample collection	Cross-sectional sampling from patients is appropriate to discover and validate diagnostic microbiome signatures. Repeated samplings of the same subject (time series or longitudinal sampling) ensure more insights into temporal dynamics and community changes. Longitudinal sampling should be chosen for monitoring disease severity or response to a treatment. Frequency should be similar between subjects.
Sample collection and storage	Storage and transit conditions are important variables in microbiome study outcomes as they impact DNA yields and quality. After collecting samples, freeze immediately. When immediate freezing is not possible, short-term refrigeration (+4 °C) is helpful. An alternative is to use stabilizing solutions. Long-term storage: currently the norm is −80 °C. Minimize freezing-thawing cycles. To this aim, it is helpful to aliquot samples before freezing.
Experimental Lab procedures	Use the same procedures and reagents throughout the study. Document everything and be consistent. If, for example, different batches of an enzyme are used, document it among the metadata. DNA extraction: This is an important source of variation and bias because of the differential resistance to lysis of microbial cells. Combine chemical and mechanical lysing procedures to capture the most accurate community composition. Contamination may significantly impact results, especially if working on low-biomass samples. It may derive from laboratory contaminants (e.g., previously produced amplicons), from reagents and commercial kits (“kitome”). It is recommendable to separate pre- and post-PCR areas and to introduce appropriate negative controls in different sample processing steps (e.g., blank extraction control: DNA-free water undergoes DNA extraction and all subsequent experimental procedures; blank PCR control: DNA-free water undergoes PCR and all subsequent procedures). Selection of 16S primers: Rely on previous studies and consider that different couples of universal 16S primers may be biased toward (or against) certain bacterial taxa, thus giving artefactual over- (or under-representations) of them. For example, the 27F/338R primer sets (targeting the V1–V3 regions) is biased against the amplification of *Bifidobacteria*. Another possible pitfall is given by primer sets poorly resolving specific taxa. PCR amplification: Low DNA template concentration and high number of PCR cycles introduce biases. To reduce their effects, minimize PCR cycles, use a standard (and relatively high) DNA template concentration, and pool multiple PCR (e.g., triplicates) for each sample. The use of proof-reading DNA polymerases and longer annealing times (to reduce chimera formation) is also recommended.
Sequencing	Use positive controls to calibrate the sequencing method: (i) pure strains of, e.g., *Escherichia coli* that produce strong PCR bands of a known size; and (ii) a synthetic mock microbial community to ensure that amplification, sequencing, and taxonomic classification workflows have not introduced substantial bias or distortions in the expected microbiome profiles. Consider that, in addition to the DNA extraction and PCR steps, errors can be introduced during library preparation, sequencing, imaging, and data analysis.
Data analysis	The design and choice of the analyses is strictly connected with the research objectives of the study. Be consistent with the procedures and software used for analyzing data. Consider that different software versions can behave differently. Integrate non-microbiome sources of data (e.g., clinical parameters) with microbiome data to answer the biological questions that primed the study. Consider that microbiota data are high-dimensional in nature, with the total number of variable measurements far exceeding the number of samples. Incorporate the patient and experimental covariates collected in the “metadata” file of the analysis. Evaluate if some of them act as confounding factors. Repeat the analyses introducing some changes (e.g., change some parameters or algorithms, include or exclude metadata) and the evaluate reproducibility of results. The complexity of questions in a translational study makes its useful to test multiple statistical models using several combinations of independent-dependent variables. If a variable is continuous, using it directly in the model is substantially more informative than using a categorical or binary encoding. Remember that DNA-based techniques are not able to reveal if the microbes under study are alive or dead. If precise information on this is needed, consider performing meta-transcriptomics.
Risk-benefit assessment	Studies need to be designed to ensure that short term and long-term reliable data are collected.

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
