# Peer review of "What Pediatricians Should Know before Studying Gut Microbiota"

_jcm, 2019, doi:10.3390/jcm8081206_

Round 1

Reviewer 1 Report

The authors review and describe current knowledge on the gut microbiota composition and analysis in early life. The manuscript gives a nice overview and provides the interested pediatrician with basic information about the gut microbiota.

To improve the manuscript, I have the following suggestions/comments:

1)    The abstract is promising more than the review actually addresses. For example, there is no section on data analyses and the recommendation of how to handle pitfalls is buried in the text. A table to summarize pitfalls and how to handle them would be helpful.

2)    The authors repeatedly use the term “microbioma”, which is not used in the English language

3)    Line 109: typo: childwood

4)    Line 114f: The authors should discuss in more detail that there is a debate whether the detected bacteria in the placenta etc are actually alive. The discovery of DNA does not mean that there is living bacteria.

5)    Figure 1: The 5 simple steps are too simple. The authors should add short descriptions to each bullet point what exactly they mean. For example, “multiple samples from the same child”, does this mean on consecutive days? Over a long time?

6)    Line 146: What is meant by “trying to avoid any internal or external interference with the final results”?

7)    Figure 2: please provide a figure legend explaining the symbols and their influence

8)    Line 197: Same as above, please discuss that there is a debate about the source of the “milk microbiota”, which in some cases originated from the mother’s breast.

9)    Line 276: From reading the description of storage/processing of the samples it seems that it does not matter how the samples are handled. This is absolutely not the case and there are several studies showing an effect of freezing/thawing and storage temperature on gut microbiota composition. This part of the manuscript should be more balanced.

10)Line 327: the authors rather refer to Figure 1.

11)As this review obviously targets pediatricians, it would be helpful if the authors would include a section with more specific, practical advices. This could be included in Figure 1, but with being more precise.

Author Response

Dear Reviewer, we would like to thank you for the suggestions, which have surely improved the manuscript. 

Best Regards

Reviewer 2 Report

What paediatricians should know before studying gut microbiota

The authors provided an overview of infant gut microbiome development and factors influencing its composition to inform pediatrics gut microbiota study design.

Major comments:

1.    Although the authors have focused on the gut microbiota, the points discussed are equally relevant to the study of the microbiome in other body parts. I suggest that the authors shift the focus from gut microbiota to the infant/child microbiome in general.

2.    Figure 1 is very simplistic and some steps are confusing. For example, what do the authors mean by “harmonising individuals according to each growth phase”?  Do they mean taking age into account as a confounder? Or what do they mean by “evaluate any interference factor to rule in or to rule out individuals”? Are they talking about confounders? For step 3, do the authors suggest to collect multiple samples at the same time, or longitudinally at multiple time points? Between steps 4 and 5, the experimental step is missing. I suggest revising your suggestions according to the current consensus and giving details of what should be done for each step as well.

3.    Line 145-146: The authors have suggested that “paediatricians should necessarily select children cohorts on the basis of their stage or phase”. I don’t agree with this suggestion. The choice of study participants will first and foremost depend on the study question. What is crucially important is to collect information about the age as well as other confounding, mediating, and moderating factors that can either influence the microbiome composition or the outcome of interest. 

4.    Line 238-239: The authors suggest to exclude any participant who has received antibiotics in the last 6 months from the study. This is not recommended in microbiome research. However, it is important to accurately document any history of antibiotics as well as other medication use.

5.    Line 258-259: similar to the point raised in comment 4, it is not recommended to consider factors that affect the microbiome composition as exclusion criteria, instead they should be carefully detailed so that in the downstream analyses they are taken into account. 

6.    Line 278: according to all guidelines and protocols, it is strongly suggested to store the samples as soon as possible at -80 °C. There are studies comparing different collection protocols and mock shipment, but sample collection is the most important and expensive part of the study, and thus the samples should be stored in a way that is suitable not only for the current project but also for any future re-analysis with newer technologies. 

7.    Line 285-287: Also, the emphasis should be to avoid freeze-thaw of the samples and thus aliquoting at the beginning. The findings of the cited study are not the consensus recommendations in general. 

Minor comments:

8.    Line 32: microbiome studies could lead to both therapeutic and preventive interventions.

9.    Line 41: please add sample collection to the list of steps to be standardised.

10.  Line 42 and 43: change microbioma to microbiome

11.  Line 42: add host immune system as another factor determining microbiome composition.

12.  Normobiota is not a commonly used term. In fact the definition of normal microbiota is still not available. Please revise.

13.  Line 44-45: it is not clear what the authors mean by “creating heterogeneous children microbiota”. Generally it is known that there is a high degree of inter-individual variability in microbiome composition both in children and adults.

14.  Line 53: since this is written for the clinician audience, define and explain trophic relationship.

15.  Line 55: highest density of microbiota in lower segments of the intestinal tract

16.  Line 56: add unit to the number of bacteria: per gram of stool

17.  Line 58: explain why fecal samples are commonly studied: non-invasive, simple

18.  Line 59-60: gut microbiome does not refer to the interaction of bacterial genome with (host) genetic factors in the mucosa. Please omit.

19.  Lines 64-66: this small paragraph seems to be out of place. Either put it in a more appropriate place or remove.

20.  Line 72: metabolomics studies all the metabolites regardless of their source (i.e. bacteria or the host). Please rephrase the last part as: produced through the interaction of diet and microbiome

21.  Line 74: please omit the 16S rRNA in parenthesis

22.  Line 83: mass spectrometry AND/OR 16S rRNA sequencing

23.  Line 84-87: this section on HMP is also out of place, either put in a more appropriate place or remove.

24.  Line 117: discuss the controversies of placenta and amniotic fluid microbiome studies especially the high likelihood of reagent contamination.

25.  Line 147: Do the authors mean confounding factors when they say “internal or external interferences”?

26.  Line 150: pilot study and previous studies would not be the best approach to sample size estimation in microbiome research. Please include some of the more recently developed methods such as the method developed for Flemish Gut Flora Project

27.  Figure 2: the title does not reflect panel a.

28.  Line 168-174: please explain and elaborate why maternal diet and lifestyle might be relevant to the infant/child gut microbiota: initial transfer of maternal microbiota, shared diet and lifestyle later in life

29.  Line 204: what do you mean by regionality?

30.  Line 204-206: It is not clear to me what this sentence means.

31.  Line 212: is in primis commonly used in English?

32.  Line 271: the authors suggest that at least 3 samples be collected. Would this be on the same day? On consecutive days? Or at different time points? Overall day-to-day variation of the gut microbiota is negligible. What is more important is sample homogenisation prior to DNA extraction.

33.  Line 272: Why are the authors suggesting to involve at least 2 sampling centers? This will add to variability. 

34.  Line 294: stool is not a low biomass sample even in infants.

35.  Line 304: Anna Karenina Principal is “based” on Tolstoy’s great book and “cited” by him.

36.  Line 311: thanks for discussing the stochastic nature of microbiome dynamic. However, the stochastic event is “drift” and not “dispersal”.

37.  Line 319-334: The focus of the paper has been on research and study design, but in this concluding paragraphs the authors switch to clinical practice and diagnosis. In my opinion, concluding on good microbiome research practices, moving from correlational studies to causation investigations, and translation would be more appropriate.

38.  English language edit is recommended. 

Author Response

(The authors gave the same response as above.)

Round 2

Reviewer 2 Report

Major comment:

Table 1 is missing

Minor comments:

Line numbers are from the tracked change document

Abstract, line 22: please change fecal/swab samples to something ore broad not focusing on the gut such as “different sampling methods” Figure 1: Improvements are good. There should be a step between 4 and 5 on the “sequencing and/or other experimental approaches” Subheading 4.1 seems unnecessary as there are no other subsections under section 4. Consider removing. Line 250: small mistake: To this regard “is it”, change to it is Figure 2: Very good improvement. In the legend please specifically refer to each panel. Also now the arrows are removed, so the description should also be removed from the legend. Line 454: consider changing diurnal oscillations to circadian rhythm disruption Line 454: are changes in latitude without time zone shift associated with circadian disruption? Line 471: how often to sample stool: consider changing to how often to collect sample Line 482: for reasons stool collection and immediate freezing might not be suitable mention large scale cohort or remote/rural areas Line 487: change indexes to indices Line 513: typo in starting

Author Response

Dear Reviewer,

                     thank you so much for your further requests and revisions.

Table 1 had been uploaded as supplementary file in the previous submission, now you may find it inside the manuscript.

Figure 1 has been further improved thanks to your suggestion, and the legenda changed accordingly (7 golden points).

Along the text you may also find all the other suggestions and corrections made.

My best regards.

Prof Lorenzo Drago